# SlotSAM: Bootstrap Segmentation Foundation Model under Real-world Shifts via Object-Centric Learning

## Abstract

Foundation models have made incredible strides in achieving zero-shot or few-shot generalization, leveraging prompt engineering to mimic the problem-solving approach of human intelligence. However, when it comes to some foundation models like Segment Anything, there is still a challenge in performing well under real-world shift. One of the real-world shift is the distribution shift, the out-of-distribution data, such as camouflaged and medical images. Another is inconsistent prompting strategies during fine-tuning and testing, leading to decreased performance. We draw inspiration from human intelligence, particularly the process by which individuals decompose scenes into components in unfamiliar environments to determine the positions or boundaries of each component. To this end, we introduce **SlotSAM**, a method that reconstructs features from the encoder in a self-supervised manner to create object-centric representations. These representations are then integrated into the foundation model, bolstering its object-level perceptual capabilities while reducing the impact of distribution-related variables. The beauty of SlotSAM lies in its simplicity and adaptability to various tasks, making it a versatile solution that significantly enhances the generalization abilities of foundation models. Through limited parameter fine-tuning in a bootstrap manner, our approach paves the way for improved generalization in novel environments.

## 1 Introduction

The impressive capabilities of foundation models (Kirillov et al., 2023; Ke et al., 2024; Radford et al., 2021; Roziere et al., 2023; Touvron et al., 2023) in zero-shot learning are a significant factor in their growing prominence. Taking the segmentation foundation models as an example, their primary goal is to achieve strong performance in dense predictions on arbitrary images, with the Segment Anything Model (SAM) (Kirillov et al., 2023) being a representative work. Despite SAM's claims robust zero-shot segmentation capabilities, distribution shift in challenging downstream tasks (e.g., medical imaging, camouflaged objects, low-quality images) undermines its advantages.

Enhancing SAM's generalization and robustness on new data is a key focus. Fine-tuning is an intuitive method to adapt SAM to various downstream tasks. This may involve customizing a medical image-specific adapter (Ma et al., 2024a)

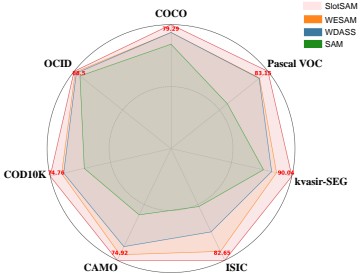

Figure 1: Performance comparison between SOTAs and SlotSAM across downstream tasks under distribution shift and prompt shift.

or integrating SAM as an additional supervisory branch (Zhang et al., 2023). However, these techniques require retraining on datasets with fine-grained annotations, often unavailable in real-world scenarios. Recent research (Li et al., 2024) has employed Stable-Diffusion to enhance a subset of the SA-1B (Kirillov et al., 2023) dataset, which requires unsustainable consumption of resources. WESAM (Zhang et al., 2024) focuses on adapting the SAM by incorporating a frozen source model under weak supervision, it utilizes LoRA (Hu et al., 2021) to fine-tune the model, thereby diminish-

ing reliance on data and computational resources. However, its enforcement of contrastive learning between different instances within images disrupts the semantic relationships among similar objects and can result in error accumulation.

The poor performance of current foundation models in unknown environments can be attributed to two types of real-world shifts. The first is **distribution shift** (Koh et al., 2021; Taori et al., 2020), which occurs when the data used for training (source domain) has a different distribution from the data encountered in downstream tasks during the actual application (target domain). The second is **prompt shift** (Abdul Samadh et al., 2024; Zhou, 2018), where downstream tasks provide only coarse weak supervision instead of the fine-grained labels available in the source domain.

To address these challenges, we draw inspiration from the perceptual pipeline of humans, particularly the process by which individuals decompose scenes into components in unfamiliar environments to determine the positions or boundaries of each component. We aim to simulate human-like intelligence (Burns et al., 2023) by abstracting the real world at the object level and injecting this capability into any foundation model. Object-centric learning (Locatello et al., 2020) operates based on causal mechanisms that align with the physical world. By leveraging its combinatorial reasoning properties in scene comprehension, object-centric learning reduces reliance on domain-specific variables and enables more robust handling of out-of-distribution data. However, applying Slot-Attention (Locatello et al., 2020), the core technology of object-centric learning, to unsupervised RGB pixel reconstruction in foundation models lacks meaningfulness for three reasons: (1) The optimization objective of reconstructing the image itself lacks sufficient information for discerning real-world objects, potentially leading to degradation, as shown in Figure 6. (2) The training of foundation models typically involves large-sized images, resulting in unacceptable resource overhead associated with Slot-Attention. (3) The injection of object-centric representations compatible with foundation models and the enhancement of their object perception capabilities warrant careful consideration.

Considering the aforementioned factors, our objective is to redefine the reconstruction target of Slot-Attention as high-level features with stronger inductive biases. Since the encoder of the foundation model effectively extracts high-level semantics for each object within the image, it offers a uniform representation of the high-dimensional nature of the real world without being biased by pixel color reconstruction. After the acquisition of high-quality object-centric representations, they could be seamlessly integrated with existing tokens in most foundation models and can be considered as object tokens. During the forward process, object tokens can leverage the attention mechanism among tokens to access global image context, geometric region, semantic information, and mask regions. This significantly enhances the foundation model's object perception capabilities with minimal fine-tuning parameters. As the entire process is unsupervised and reinforces the generalizability of the foundation model, relying on its exceptional feature representation, we define it as bootstrapping. Our contributions can be summarized as follows:

- We propose SlotSAM to obtain high-quality representations from foundation model and innovatively project them as object tokens that can be seamlessly integrated.

- We utilize object tokens and attention mechanisms to access the global context, geometry, and semantics of the image, injecting object-centric perceptual ability into the foundation model with only minimal parameter adjustments.

- In real-world scenarios, including distribution shift and prompt shift, SlotSAM significantly enhances segmentation accuracy across various downstream tasks, facilitating the safe deployment of the foundation model in the open world.

## 2 PRELIMINARIES

We intend to provide a way to inject object-centric representation perception capabilities into foundation models in a general sense. Therefore, the training process of the foundation models is not the focus of attention, so we do not differentiate between the optimization objectives of the original foundation models or the fine-tuned foundation models, modeling their loss functions as $\mathcal{L}_{\text{base}}$.

We chose the SAM as a representative foundation model for our research. SAM consists of three main components: the image encoder $\mathbf{z} = f(\mathbf{x}; \Theta)$, the prompt encoder $\mathbf{e} = g(\mathbf{p}; \Omega)$, and the mask

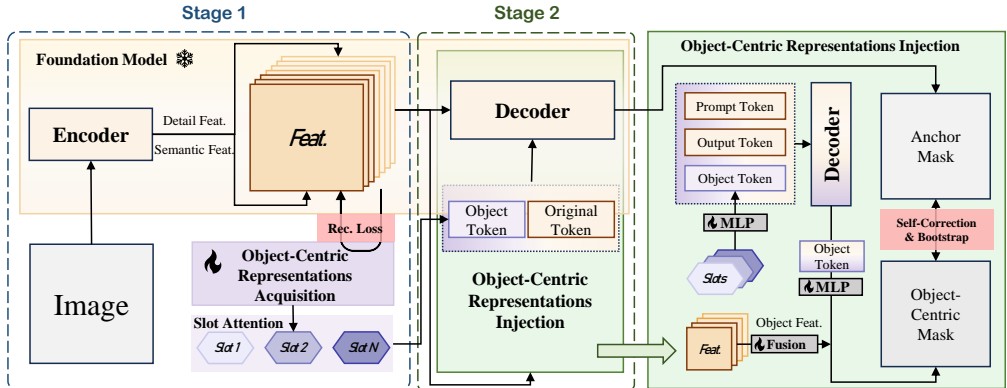

Figure 2: Overview of SlotSAM. Obtaining slots by reconstructing higher-order semantics at stage 1. Injecting slots into the foundation model by nonlinearly combining them into object token and self-training at stage 2. The whole process is task-agnostic.

decoder $h(\mathbf{z}, \mathbf{e}; \Phi)$. For SAM and any improvements made to SAM, we generally represent their optimization objectives as $\mathcal{L}_{\text{base}}$. In this paper, the $\mathcal{L}_{\text{base}}$ is derived from the WESAM (Zhang et al., 2024). For each input $\mathbf{x}$, we obtain $\mathbf{x}_s$ and $\mathbf{x}_w$ through strong augmentation and weak augmentation, respectively. $\mathbf{x}_w$ is then processed by the anchor model $f(\mathbf{x}_w; \Theta^a)$ and the student model $f(\mathbf{x}_w; \Theta^s)$ to obtain $\mathcal{M}^a$ and $\mathcal{M}^s$, while the teacher model $f(\mathbf{x}_s; \Theta^t)$ processes $\mathbf{x}_w$ to obtain $\mathcal{M}^t$. $\mathcal{M}$ represents the predicted mask. A generic and base self-training loss $\mathcal{L}_{\text{base}}$ can be defined as:

$$\mathcal{L}_{\text{base}} = \mathcal{L}^{\text{dice}}\left(\mathcal{M}^{s/t}, \mathcal{M}^a\right) + \mathcal{L}^{\text{focal}}\left(\mathcal{M}^s, \mathcal{M}^t\right). \tag{1}$$

## 3 METHODOLOGY

### 3.1 OBJECT-CENTRIC REPRESENTATION ACQUISITION

Simply reconstructing RGB pixels allows Slot-Attention to achieve some effectiveness on synthetic datasets, but in the real world, RGB supervision signals are insufficient to represent objects and environments, making them prone to degradation, as shown in Figure 6. Inspired by (Seitzer et al., 2023), object-centric representation requires a more well-trained semantic encoder, and fortunately, the encoder of the foundation model can provide rich semantic details.

The underlying logic of Slot-Attention is to reconstruct features through self-supervision, compressing high-dimensional, semantically rich, and unstructured object features into low-dimensional structured information in a bottleneck-like manner. Slots act as the bottleneck, retaining object-centric representation. Therefore, given the output feature $\mathbf{z} \in \mathbb{R}^{N \times D_z}$ from the encoder $f(\mathbf{x}; \Theta)$, and initializing a set of slots $\mathbf{s} \sim \mathcal{N}(\mathbf{s}; \boldsymbol{\mu}, \boldsymbol{\sigma}) \in \mathbb{R}^{K \times D_s}$, $K$ is the number of slots, $D_z$ and $D_s$ represent the dimension of output feature and slot. We project them to the dimension by a linear transformation $\mathcal{K}_\beta$ for slots and $\mathcal{Q}_\gamma, \mathcal{V}_\phi$ for $\mathbf{z}$, and the Slot-Attention is trained as $\text{update}(\boldsymbol{A}, \mathbf{v}) = \boldsymbol{A}^T \mathbf{v}$, where $\text{update}(\boldsymbol{A}, \mathbf{v}) = \boldsymbol{A}^T \mathbf{v}, A_{ij} = \frac{\text{attn}(\mathbf{q},\mathbf{k})_{ij}}{\sum_{l=1}^K \text{attn}(\mathbf{q},\mathbf{k})_{lj}}, \text{attn}(\mathbf{q}, \mathbf{k}) = \frac{e^{M_{ij}}}{\sum_{l=1}^N e^{M_{il}}}, \boldsymbol{M} = \frac{\mathbf{k}\mathbf{q}^T}{\sqrt{D_s}}$. The $\mathbf{q} = \mathcal{Q}_\gamma(\mathbf{z}) \in \mathbb{R}^{K \times D_s}, \mathbf{k} = \mathcal{K}_\beta(\mathbf{z}) \in \mathbb{R}^{N \times D_s}$, and $\mathbf{v} = \mathcal{V}_\phi(\mathbf{z}) \in \mathbb{R}^{N \times D_s}$ denote the query, key and value vectors respectively, and the query is a function of the slots. After optimizing $T$ iterations using the Gated Recurrent Unit (Chung et al., 2014; Dey & Salem, 2017) (GRU), the slots are passed through a slot-decoder to output the reconstructed feature $\hat{\mathbf{z}}$, minimizing the self-supervised reconstruction loss:

$$\mathcal{L}_{\text{rec}} = \|\hat{\mathbf{z}} - \mathbf{z}\|^2, \quad \hat{\mathbf{z}} = \text{slot-decoder}(\mathbf{s}). \tag{2}$$

$\hat{\mathbf{z}}$ is the weighted sum of each slots. Since each slot should be associated with a different object, each slot should be able to attend to specific spatial regions. Following (Seitzer et al., 2023), we employ

an efficient MLP as a spatial broadcast decoder (Watters et al., 2019). Each slot is broadcasted to several patches with the addition of positional encoding. The tokens for each slot are processed individually by the MLP, and after channel division, we obtain the reconstructed feature $\hat{\mathbf{z}}_k$ and the activation region $\alpha_k$. The weighted feature representation $\hat{\mathbf{z}}$ for all slots after reconstruction is obtained by

$$\hat{\mathbf{z}} = \sum_{k=1}^{K} \hat{\mathbf{z}}_k \odot \boldsymbol{m}_k, \quad \boldsymbol{m}_k = \operatorname*{softmax}_k(\alpha_k). \tag{3}$$

## 3.2 OBJECT-CENTRIC REPRESENTATION INJECTION

In the decoder of SAM, the predicted mask is obtained by performing element-wise multiplication between the Output Token $\mathcal{T}_{out} \in \mathbb{R}^{N_{out} \times D_{out}}$ and the mask feature. The accuracy of the mask is strongly correlated with the amount of information provided by the tokens. Therefore, as shown in Figure 2, we innovatively design the object-centric representation stored in the slots to be the Object Token. This design is fully compatible with the original decoder architecture, and thanks to the attention mechanism, the Object Token can exchange information with other tokens. The Object Token can access the global image's contextual information and geometric details. Furthermore, the existing $\mathcal{T}_{out}$ can acquire more discriminative features related to objects, such as positional information and topological associations.

For each input $\mathbf{x}$, there is a corresponding set of slots $\mathbf{s}$. To avoid disrupting the optimization preference established by the decoder for existing tokens, $\mathbf{s} \in \mathbb{R}^{K \times D_s}$ is fed into an MLP for nonlinear combination to obtain the Object Token $\mathcal{T}_{obj} \in \mathbb{R}^{N_o \times D_s}$, where $D_s = D_{out}$. In each attention layer, the Object Token performs self-attention calculations with other tokens and shares the same feed-forward layers to ensure consistent optimization direction of model.

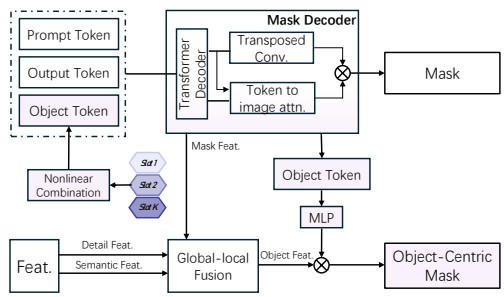

Figure 3: HQSAM-style object-centric decoder.

As the $\mathcal{T}_{obj}$ contains more deep semantic features and fewer detailed features, introducing local boundary details helps avoid boundary blurring for objects. We introduce an HQSAM-style (Ke et al., 2024) object-centric decoder (Figure 3) for the injection of $\mathcal{T}_{obj}$. We extract the detail features from the first attention block of the encoder and apply transposed convolution. After that, we add the detail feature with the semantic features to obtain the fused object features $\mathbf{z}^{obj}$. Then, similar operations are performed, where the $\mathcal{T}_{obj}$ is multiplied by the $\mathbf{z}^{obj}$ to obtain the Object-Centric Mask $\mathcal{M}^o$.

Self-training networks may suffer from the problem of error accumulation due to incorrect predictions. Therefore, in the early stages of training, we fix the parameters of the anchor model (with $\mathbf{x}_w$ as the input). The trained model is referred to as the object-centric model (with $\mathbf{x}_s$ as the input). We use a simplified loss function in the style of $\mathcal{L}_{\text{base}}$ to train the MLP and Fusion modules, in order to prevent significant bias in knowledge transfer:

$$\mathcal{L}^{\text{dice}}\left(\mathcal{M}^o, \mathcal{M}^a\right) + \mathcal{L}^{\text{bce}}\left(\mathcal{M}^o, \mathcal{M}^a\right). \tag{4}$$

In the later stages of training, we employ a bootstrap strategy. At the end of epochs where the model has improved its mIoU on the validation set, we directly copy the parameters of the object-centric model to the anchor model. Through this iterative process, we gradually complete the bootstrap of the foundation model.

Table 1: Comparison with SOTAs on natural and medical image datasets using bounding box , sparse points , and coarse segmentation mask prompts. † denotes reproduced results.

| Method | COCO 2017 | | | Pascal VOC | | | kvasir-SEG | | | ISIC | | |
|---|---|---|---|---|---|---|---|---|---|---|---|---|
| | box | point | poly | box | point | poly | box | point | poly | box | point | poly |
| SAM (Kirillov et al., 2023) | 74.29 | 55.06 | 65.64 | 69.21 | 69.21 | 60.79 | 81.59 | 62.30 | 54.03 | 66.74 | 53.42 | 62.82 |
| TENT (Wang et al., 2021) | 78.21 | 52.99 | 71.51 | 80.24 | 74.97 | 65.03 | 82.47 | 61.84 | 62.97 | 71.76 | 53.46 | 67.12 |
| SHOT (Liang et al., 2021) | 75.18 | 58.46 | 69.26 | 79.80 | 74.26 | 63.38 | 82.30 | 63.76 | 61.34 | 71.99 | 55.99 | 66.86 |
| Soft Teacher (Xu et al., 2021) | 75.94 | 43.36 | 68.27 | 72.93 | 56.09 | 62.20 | 84.12 | 73.53 | 58.15 | 75.74 | 54.95 | 72.29 |
| TRIBE (Su et al., 2024) | 77.56 | 49.56 | 70.99 | 78.87 | 69.21 | 65.39 | 85.05 | 73.03 | 64.61 | 72.61 | 50.36 | 67.99 |
| DePT (Gao et al., 2022) | 71.00 | 37.35 | 63.27 | 74.09 | 42.99 | 59.94 | 81.91 | 52.06 | 61.55 | 78.43 | 46.79 | 72.75 |
| WDASS (Das et al., 2023) | 77.29 | 60.55 | 70.19 | 80.12 | 76.15 | 66.98 | 84.01 | 63.78 | 64.78 | 74.23 | 55.63 | 67.84 |
| WESAM† (Zhang et al., 2024) | 77.32 | 60.5 | 70.77 | 80.27 | 74.15 | 66.72 | 85.47 | 75.23 | 67.40 | 80.01 | 62.12 | 75.36 |
| Ours | **79.29** | **60.99** | **75.48** | **83.15** | **77.23** | **70.77** | **90.04** | **81.96** | **79.64** | **82.65** | **66.21** | **78.72** |
| △ | +5.00 | +5.93 | +9.84 | +13.94 | +8.02 | +9.98 | +8.45 | +19.66 | +25.61 | +15.91 | +12.79 | +15.90 |
| Supervised | 81.50 | 69.77 | 73.39 | 81.23 | 76.98 | 71.32 | 85.89 | 77.54 | 81.64 | 81.62 | 79.81 | 80.26 |

# 4 EXPERIMENTS

## 4.1 SETUP

**Benchmarks.** We conduct evaluations under real-world shifts (distribution and prompt shift).

*Distribution shift*: The training dataset for SAM from the source domain is SAIB, primarily collected from natural environments. We select four types of real-world downstream task datasets as the target domain. Among them, the distribution shift is small for the natural image datasets VOC (Ke et al., 2024) and COCO (Ke et al., 2024), and the robotic image dataset OCID (Ke et al., 2024). In contrast, the distribution shift is significant for the medical dataset ISIC (Ke et al., 2024), PolyP (Ke et al., 2024), and the camouflaged object image dataset CAMO (Ke et al., 2024). The division of the training and test sets follows previous work (Das et al., 2023; Chen et al., 2023; Zhang et al., 2024), with fine-tuning performed on the training set and the test set used for performance evaluation.

*Prompt shift*: Fine-tuning is weakly supervised, meaning that the available labels are incomplete, which is different from the fine mask labels used in the training of the foundation model and is considered as prompt shift. Consistent with existing work, we obtain the minimal bounding box that completely encompasses the instance segmentation mask, which serves as the box prompt. The point prompt is generated by randomly selecting five positive points within the ground-truth instance segmentation mask and five negative points outside it. A coarse segmentation mask is simulated by fitting a polygon around the ground-truth mask, where the number of vertices is determined as $P/20$, with $P$ representing the mask's perimeter. The minimum number of vertices required is three.

**Evaluation Protocols.** We present the mean Intersection over Union (mIoU) as the primary evaluation metric. For each input prompt, the mIoU is calculated by comparing the ground-truth segmentation mask with the predicted mask. The final mIoU is obtained by averaging the IoU values across all instances.

**Implementation Details.** We utilize the ViT-B (Alexey, 2020) as the image encoder of SAM. The standard prompt encoderis employed in SlotSAM. Throughout all experiments, the LoRA module of the image encoder and MLP are fine-tuned using the Adam optimizer. We configure a batch size of 4 on four NVIDIA RTX4090 GPUs and set the learning rate to 0.0001 along with a weight decay of 0.0001. The rank of the LoRA module is 4. The number of epochs for fine-tuning with $\mathcal{L}_{base}$ is consistent with previous work (Zhang et al., 2024), with 400 epochs used for slots acquisition, and 20 epochs for slots injection. We set the number of object tokens $N_o$ and the number of slots $K$ to 1 and 8 respectively.

## 4.2 BASELINES.

**TENT** (Wang et al., 2021) is a simple test-time adaptation approach that optimizes entropy loss for adapting to the target domain. **SHOT** (Liang et al., 2021) employs pseudo-labels and operates under the assumption of a uniform distribution for source-free domain adaptation. **Soft Teacher** (Xu et al., 2021) initially developed for semi-supervised segmentation, has been adapted for domain adaptation with self-training. **TRIBE** (Su et al., 2024) establishes a strong baseline for generic test-time adap-

Table 2: Comparison with SOTAs on camouflaged object and robotic image datasets using bounding box , sparse points , and coarse segmentation mask prompts. † denotes reproduced results.

| Method | CAMO | | | COD10K | | | OCID | | |
|---|---|---|---|---|---|---|---|---|---|
| | box | point | poly | box | point | poly | box | point | poly |
| SAM (Kirillov et al., 2023) | 62.72 | 57.43 | 50.85 | 66.32 | 63.61 | 40.04 | 86.35 | 71.41 | 72.81 |
| TENT (Wang et al., 2021) | 71.24 | 59.59 | 60.29 | 69.36 | 61.94 | 43.36 | 87.77 | 66.61 | 77.53 |
| SHOT (Liang et al., 2021) | 71.61 | 62.78 | 58.72 | 69.09 | 65.25 | 42.38 | 88.06 | 74.39 | 76.25 |
| Soft Teacher (Xu et al., 2021) | 62.30 | 48.64 | 51.26 | 66.32 | 50.04 | 32.27 | 84.98 | 68.46 | 73.75 |
| TRIBE (Su et al., 2024) | 66.00 | 61.97 | 60.54 | 67.84 | 63.62 | 42.75 | 86.77 | 67.86 | 76.50 |
| DePT (Gao et al., 2022) | 55.44 | 33.07 | 48.63 | 59.32 | 34.06 | 35.51 | 82.00 | 56.52 | 70.92 |
| WDASS (Das et al., 2023) | 71.25 | 63.39 | 62.29 | 71.42 | 65.61 | 43.93 | 87.68 | 77.13 | 76.70 |
| WESAM† (Zhang et al., 2024) | 73.42 | 65.55 | 62.90 | 71.93 | 70.55 | 45.87 | 88.09 | 80.14 | 77.41 |
| Ours | **74.92** | **68.95** | **71.09** | **74.76** | **72.46** | **48.86** | **88.50** | **81.35** | **86.54** |
| △ | +12.20 | +11.52 | +20.24 | +8.44 | +8.85 | +8.82 | +2.15 | +9.94 | +13.73 |
| Supervised | 79.17 | 77.01 | 67.12 | 78.06 | 78.44 | 64.90 | 91.24 | 89.22 | 79.23 |

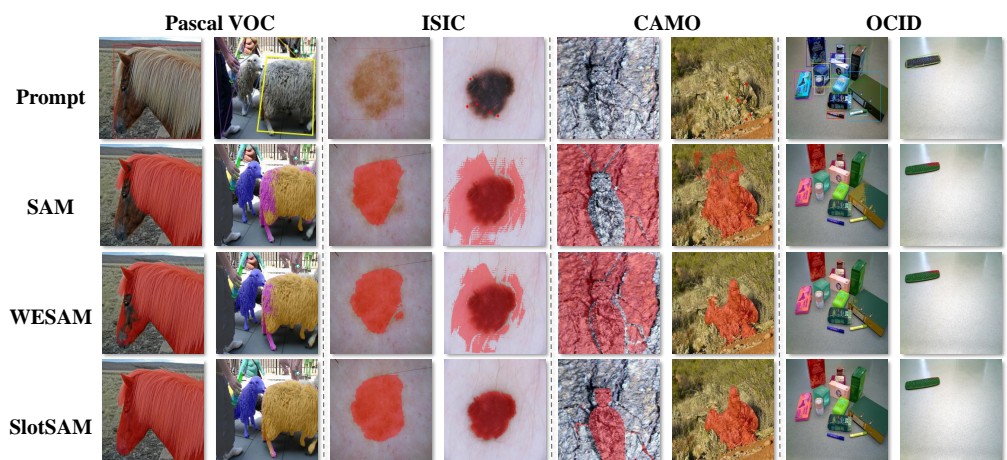

Figure 4: Comparison between SlotSAM and SOTAs of the fineness of the predicted masks.

tation in the context of continuous and class-imbalanced domain shifts. **DePT** (Gao et al., 2022) integrates visual prompts into a visual Transformer, adapting these source-initialized prompts solely during the adaptation phase without source data. **WDASS** (Das et al., 2023) introduces an approach for weakly supervised domain adaptive segmentation. **WESAM** (Zhang et al., 2024) is a domain adaptation segmentation method based on a foundation model. It utilizes teacher-student networks and instance contrastive for weakly supervised learning. Additionally, we evaluate the performance of adapting SAM through fine-tuning with ground-truth masks, referred to as Supervised.

## 4.3 RESULTS

**Quantitative results.** As shown in Table 1 and Table 2, we evaluate SlotSAM on datasets from seven real-world downstream tasks and three types of prompt methods. SlotSAM achieves comprehensive improvements on out-of-distribution datasets across four categories (natural, medical, camouflaged object, robotic), significantly outperforming existing methods. On natural images, SlotSAM significantly narrows the gap with fully supervised fine-tuning schemes, and even under the challenging prompt shift conditions, such as fine-tuning with point or poly prompts, its mIoU surpasses that of fine-grained mask supervised fine-tuning. In medical imaging, SlotSAM's mIoU on the Kvasir-SEG dataset exceeds 90%, and it continued to perform well under poly supervision, surpassing WESAM (Zhang et al., 2024) by 18.16%. Even on the most difficult camouflaged object data, we achieve an average improvement of over 3%. On robotic images, SlotSAM's advantages were evident, overcoming the limitation that the imprecise weak supervision labels provided by prompt shift did not yield significant performance gains. In summary, SlotSAM enhances the foundation model

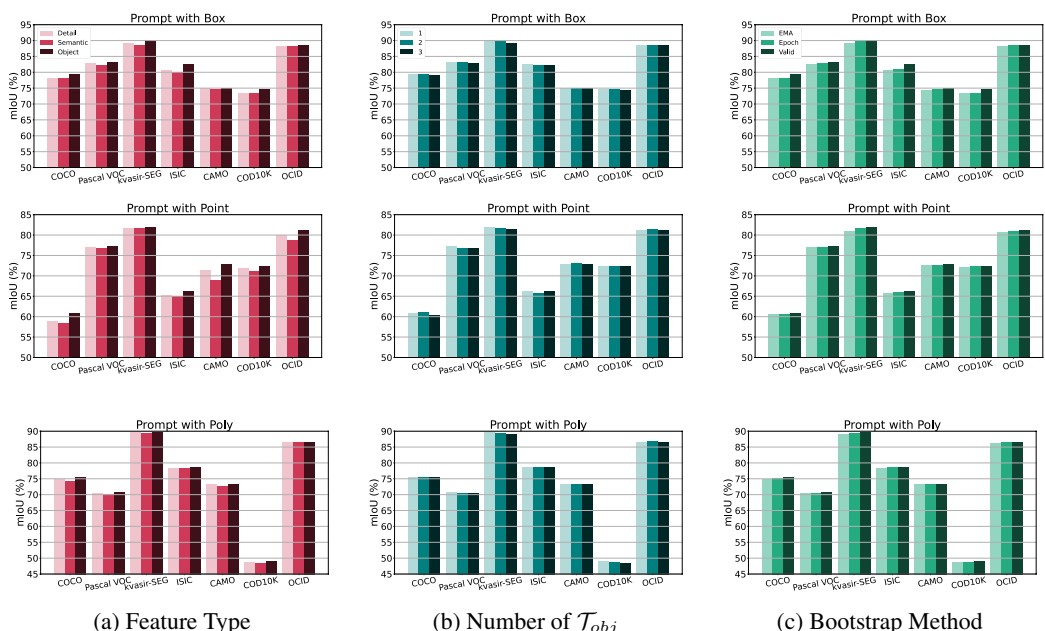

(a) Feature Type      (b) Number of $\mathcal{T}_{obj}$      (c) Bootstrap Method

Figure 5: Ablation of the three core components of SlotSAM on seven benchmarks.

with object perception capabilities, allowing it to be independent of detailed annotations. Even with only point or poly annotations provided, it can still accurately capture the location and semantic information of objects, thereby consistently making precise dense predictions in real-world scenarios.

**Qualitative results.** As shown in Figure 4, we visualize the masks predicted by SlotSAM and state-of-the-art methods. SlotSAM has two major advantages: (1) In parts with a small pixel area occupation, such as the junction of a horse's hair and face. SlotSAM can provide the most refined predictions thanks to its ability to capture objects'

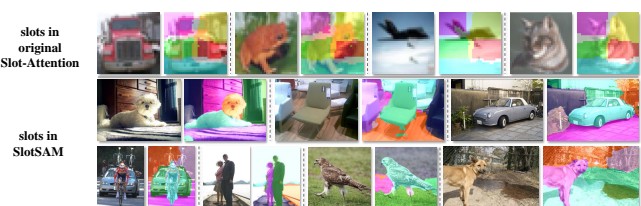

Figure 6: Comparison of the quality of the slots.

semantic correlation. The object tokens derived from the nonlinear combination of slots contain semantic information and spatial representation. Since the embeddings of the object's integrity also include the object's fine boundaries within the slots, it prevents any part of the object from being overlooked. (2) SlotSAM can provide semantic boundaries with higher distinguishability in boundary areas prone to confusion, avoiding semantic confusion. This indicates that the high semantic distinctiveness between different slots is provided to the segmentation model, enabling it to easily distinguish between different objects or the boundaries between the foreground and background. Figure 6 illustrates that SlotSAM obtains non-degenerate object-centric representations compared to the RGB-level original Slot-Attention. These representations are full-ranked, capturing the complete spectrum of visual information without loss. The focus on object-centric representations allows SlotSAM to understand the spatial relationships within a scene, resulting in segmentations that are not only accurate but also semantically meaningful.

## 4.4 ABLATION STUDIES AND ANALYSIS

**Ablations of key components in SlotSAM.** We conduct ablation experiments on the three core aspects of SlotSAM: the source of features used for merging with the object token, the number of object tokens, and the bootstrap method. The trends are summarized as follows. (1) As shown in Figure 5a, for any prompt shift, as expected, the merged object features give SlotSAM the highest

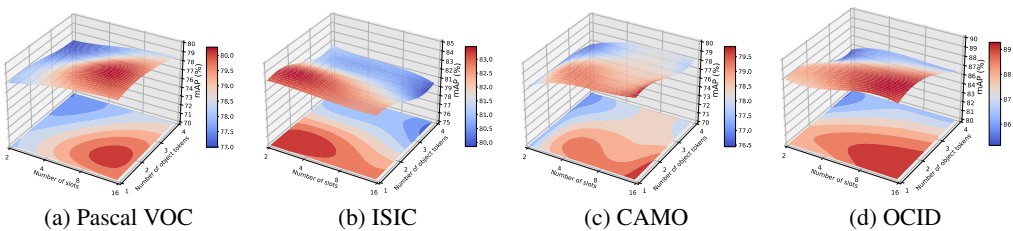

|                |              |          |            |           |
| -------------- | ------------ | -------- | ---------- | --------- |
| (a) Pascal VOC | (b) ISIC     | (c) CAMO | (d) OCID   |           |

Figure 7: The correlation between the number of object tokens $N_o$ and the number of slots $K$.

mIoU across all datasets. Moreover, the Detail feature provides the object token with more geometry and edge information related to the segmentation task. Compared to the fusion with deep semantic features, the highly compressed object token requires more complementary information. (2) Since the original prompt tokens and output tokens in the decoder are few, the number of object tokens should not be excessive. Keeping them at 1 or 2 enhances the segmentation capability for instances. The reason is that too many object tokens introduce a large number of task-irrelevant gradients during the attention between tokens. Fine-tuning with only a small amount of data can bias the model parameters, leading to catastrophic forgetting. Additionally, it can be observed in Figure 5b that complex real-world scenarios, such as COCO, require more object tokens for representation. (3) We consider three parameter update methods for the anchor model during the fine-tuning process: full parameter copy every 5 epochs, Exponential Moving Average (EMA) update, and full parameter copy when the validation set performance increases. As shown in Figure 5c, EMA has the lowest performance gain, while the Valid method is the most effective. The reason is that the anchor model requires substantial parameter updates to perceive object-centric representation and adapt to new distributions quickly.

**Analysis on the number of object tokens.** An important factor in SlotSAM is the number of object tokens, which is closely related to the number of slots. We further investigate the effects of their interaction on four types of downstream tasks. As shown in Figure 7, for complex real-world scenarios, such as the COCO, CAMO, and OCID datasets, a consistent trend is that an appropriate number of object tokens combined with as many slots as possible helps the model segment better. This is intuitive because a real-world scene contains multiple objects that require a large number of slots to correspond to, and due to the complexity of the background and environment, the object token needs redundancy to represent the image fully. Conversely, for medical images with a single instance like ISIC, a small amount of representation related to the object can fully express the entire foreground and background.

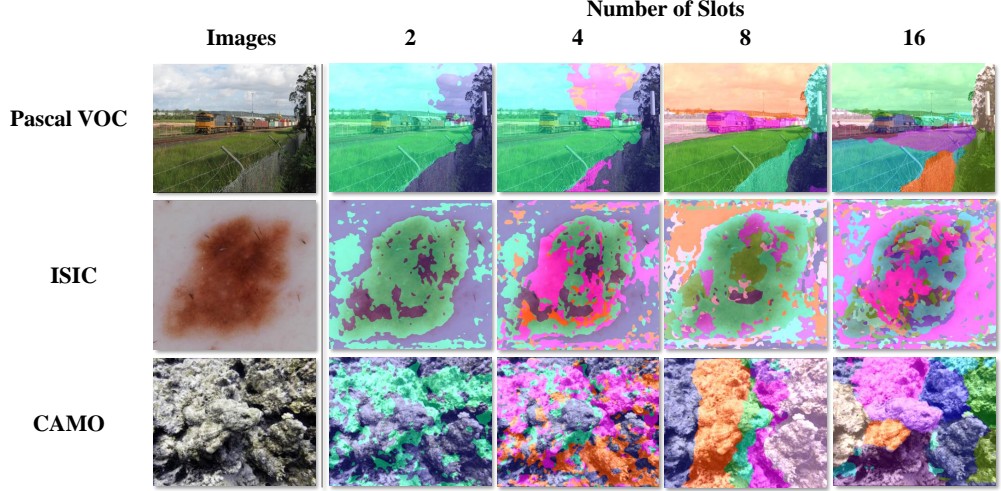

Figure 8: Semantic competition and semantic degradation exists among different numbers of slots.

**Analysis on the number of slots.** Slots correspond directly to a specific area within an image, and this area typically possesses a specific semantic meaning. The number of slots should be positively correlated with the scene's complexity and objects in the image. To explore whether there is a certain trend in the number of slots in representing objects, we visualize in Figure 8 the correspondence of areas with different numbers of slots on images from three downstream tasks. Some interesting findings are observed: (1) On natural images of the real world, both 8 and 16 slots found semantic affiliations (fewer slots focus on the entirety of grass or a train, while more slots segment the train into individual carriages). This is similar to the human process of perceiving new scenes, where one can overview various objects and further break them down into their components. (2) medical images with only a single object exhibit a competitive phenomenon when the number of slots is high. The area focused on by a single slot becomes trivial, and the competition among slots leads them to pay less attention to semantic information and more to low-dimensional patches, textures, and other information. (3) In complex environments, discovering camouflaged targets is challenging. A small number of 2 or 4 slots is insufficient to represent the entire image, and the areas focused on by the slots are discrete and lack specific semantics. The situation slightly improves when there are 8 slots. Still, there is a trend of degradation in the representation of slots, meaning that the masks of slots are related to fixed spatial unknowns rather than semantics. A surprising phenomenon occurs under 16 slots when complex scenes can be fully represented: slots can even directly discover camouflaged objects and have an excellent understanding of background areas.

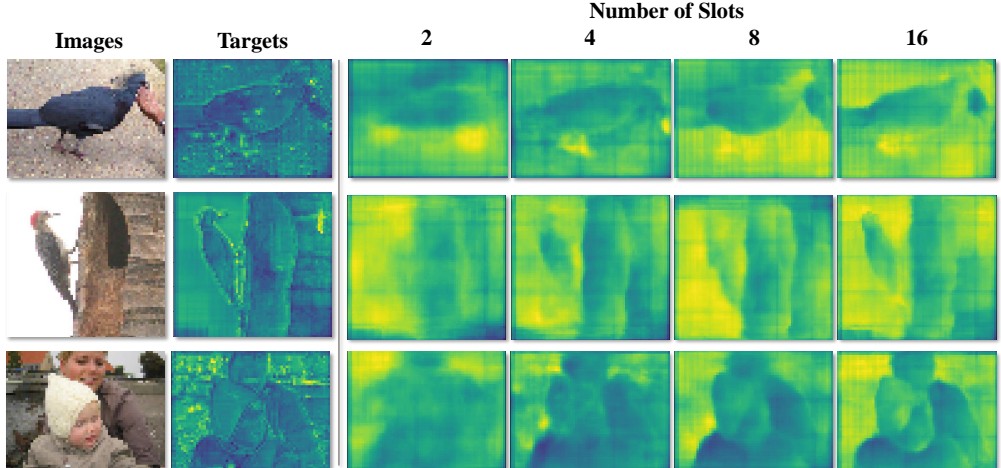

Figure 9: The preference for reconstructing details in object areas with different numbers of slots.

SlotSAM's excellent object perception capability stems from high-quality slots, which are trained using the output features of the image encoder as reconstruction targets. We visualize the reconstruction results under different numbers of slots, as shown in Figure 9. As the number of slots increases, the details of the reconstruction results become more affluent, and the slots gradually transition from representing objects to representing parts. However, it is interesting that the slots' good perception of semantics does not rely on faithfully reconstructing every feature pixel but instead selectively reconstructing (the reconstruction details of a human's hat and face are significantly more than those of the background). This suggests that some hidden reasoning mechanisms may be similar to human "focusing" when observing a new environment, directing more attention to task-relevant features.

## 5 RELATED WORK

### 5.1 SEGMENTATION FOUNDATION MODEL

In the progression of Segmentation segmentation Models (SFMs) research, the Segment Anything Model (SAM) (Kirillov et al., 2023) establishes a foundation for subsequent works, showcasing the potential of deep learning in versatile segmentation tasks. This paves the way for MedSAM (Ma et al., 2024b) to further specialize SAM for the intricate domain of medical imaging, thereby expanding its applicability to high-stakes clinical diagnostics. Moreover, the challenge of limited

annotated data in part segmentation has been addressed by the weakly-supervised part segmentation (WPS-SAM) (Wu et al., 2024), which ingeniously harnesses the pre-trained knowledge within SAM to perform segmentation with weak labels, thus complementing the strengths of SAM in scenarios with limited supervision. Furthermore, the innovative integration of 3D segmentation with neural radiance dields by SA3D (Cen et al., 2023) represents a significant leap forward, extending the reach of SAM from 2D to 3D spaces and thus providing a more comprehensive solution for volumetric data segmentation. The empirical evaluation of few-shot learning within SFMs (Chang et al., 2024) revealing their potential to adapt to new tasks with limited data. In the realm of practical applications, MedFMC (Wang et al., 2023) rigorously assesses the adaptability of SFMs to medical image classification tasks. Some research (Ma & Wang, 2023) emphasize the importance of developing SFMs that are adept at handling the complexities of biological imaging modalities, which is a significant step towards creating more specialized tools for biological research and diagnostics. Concurrently, WESAM (Zhang et al., 2024) focuses on enhancing the robustness of segmentation models under distribution shifts through weakly supervised adaptation, which is essential for ensuring the reliability of models. Our research not only emphasizes the application of the SAM in a singular context but also takes a comprehensive view of real-world scenarios. SFMs must address both distribution shifts and prompt shifts, indicating that the data used for fine-tuning the model and the format of the labels are not available during the training phase. This scenario reflects the complexities inherent in open-world settings.

## 5.2 Object-Centric Learning

Object-Centric Learning is revolutionizing the field of computer vision with its ability to perceive and understand scenes in terms of discrete objects. Slot-Attention (Locatello et al., 2020) introduces the concept of "slots" that can bind to any object in the input through a competitive attention process. It laid the foundation for subsequent innovations in object-centric representation learning. SlotLifter (Liu et al., 2024) addresses the challenge of 3D scene understanding, it aggregates multi-view features for decoding via slot-guided feature lifting, significantly advancing in scene decomposition and novel-view synthesis. An advancement (Jia et al., 2022) proposes a model that utilizes learnable queries to initialize Slot-Attention learning, demonstrating potential for concept binding and generalization. Some works (Elsayed et al., 2022; Zhao et al., 2023) focus on object tracking, introducing an index merge module and a memory module to address the challenges of object fragmentation and temporal consistency. An interesting approach (Wang et al., 2024) offers a novel perspective, using cyclic walks between parts of an object to enhance object-centric learning, providing a new way to handle object fragments and their reassembly. EAGLE (Kim et al., 2024) introduces a method for unsupervised semantic segmentation that learns object-centric representations without manual annotations, significantly reducing the reliance on annotated data. SlotSAM differs from existing research that solely investigates whether slots can effectively represent objects. We examine how foundation models perceive and interpret visual data, employing cross-distribution invariant object representations to enhance model generalization performance and facilitate secure deployment in open-world environments.

## 6 Conclusion

In this work, we introduce SlotSAM, a novel approach to enhance the generalization of segmentation foundation models under distribution and prompt shifts. Drawing inspiration from human object-centric perception, SlotSAM employs self-supervised learning to extract and integrate high-quality object-centric representations, bolstering the model's ability to perceive and segment objects in varied environments. Our method's effectiveness is underscored by its superior performance over existing state-of-the-art methods across diverse downstream tasks. The simplicity and versatility of SlotSAM make it readily integrated into various foundation models, marking a significant stride in enhancing their generalization capabilities. SlotSAM exemplifies the potential of human-inspired learning to advance machine perception, marking an important step toward developing models capable of safely and effectively navigating open-world scenarios with limited supervision.

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
