# SlotSAM: Bootstrap Segmentation Foundation Model under Real-world Shifts via Object-Centric Learning

# (Supplemental Material)

This document provides more details of our approach and additional experimental results, organized as follows:

## 1 Predicted Masks

## 2 Predicted Masks

## 3 Predicted Masks

## 4 Predicted Masks

## 5 Visualization on Slots

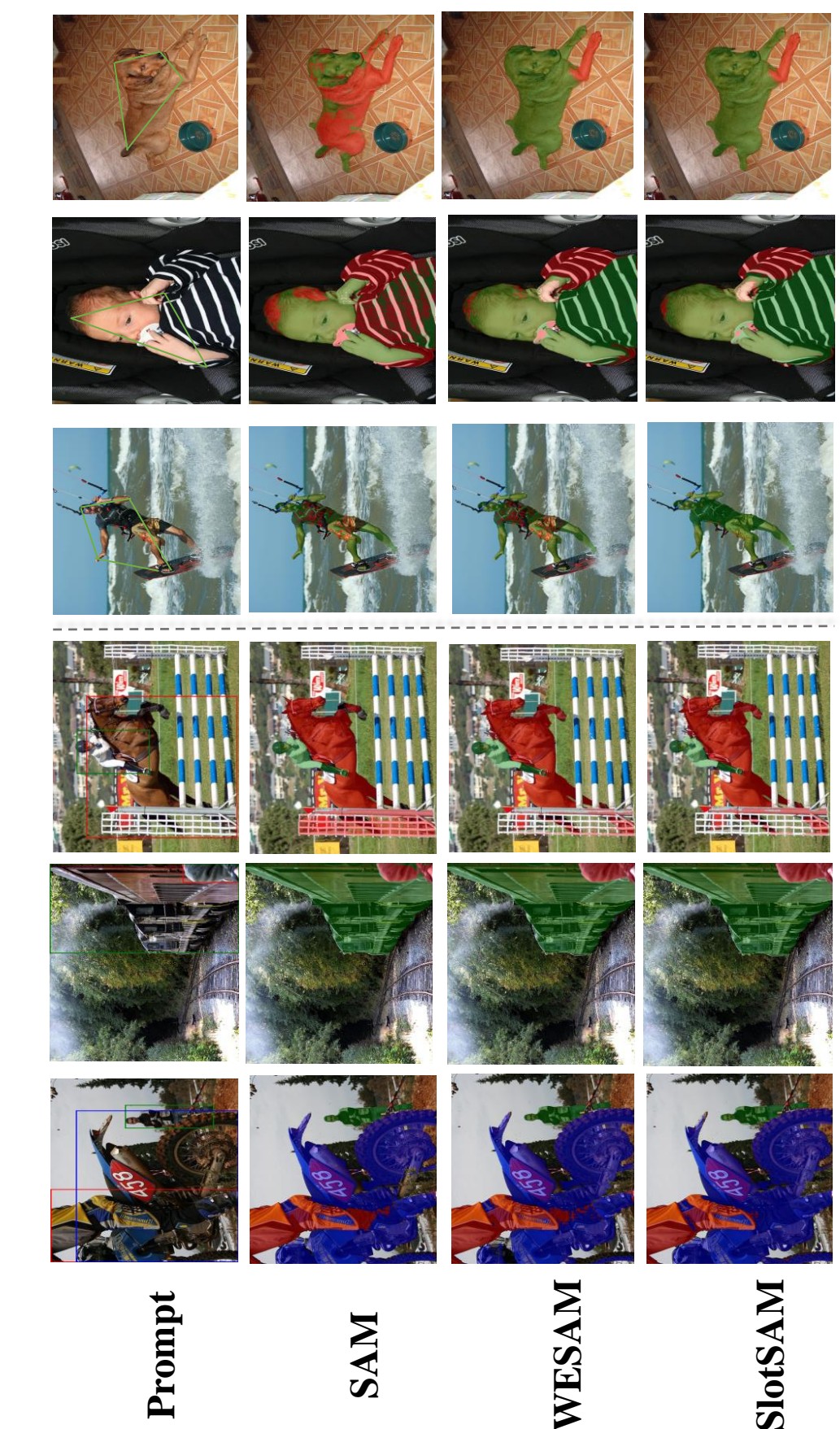

Figure 1: Comparison between SlotSAM and SOTAs of the fineness of the predicted masks on Pascal VOC.

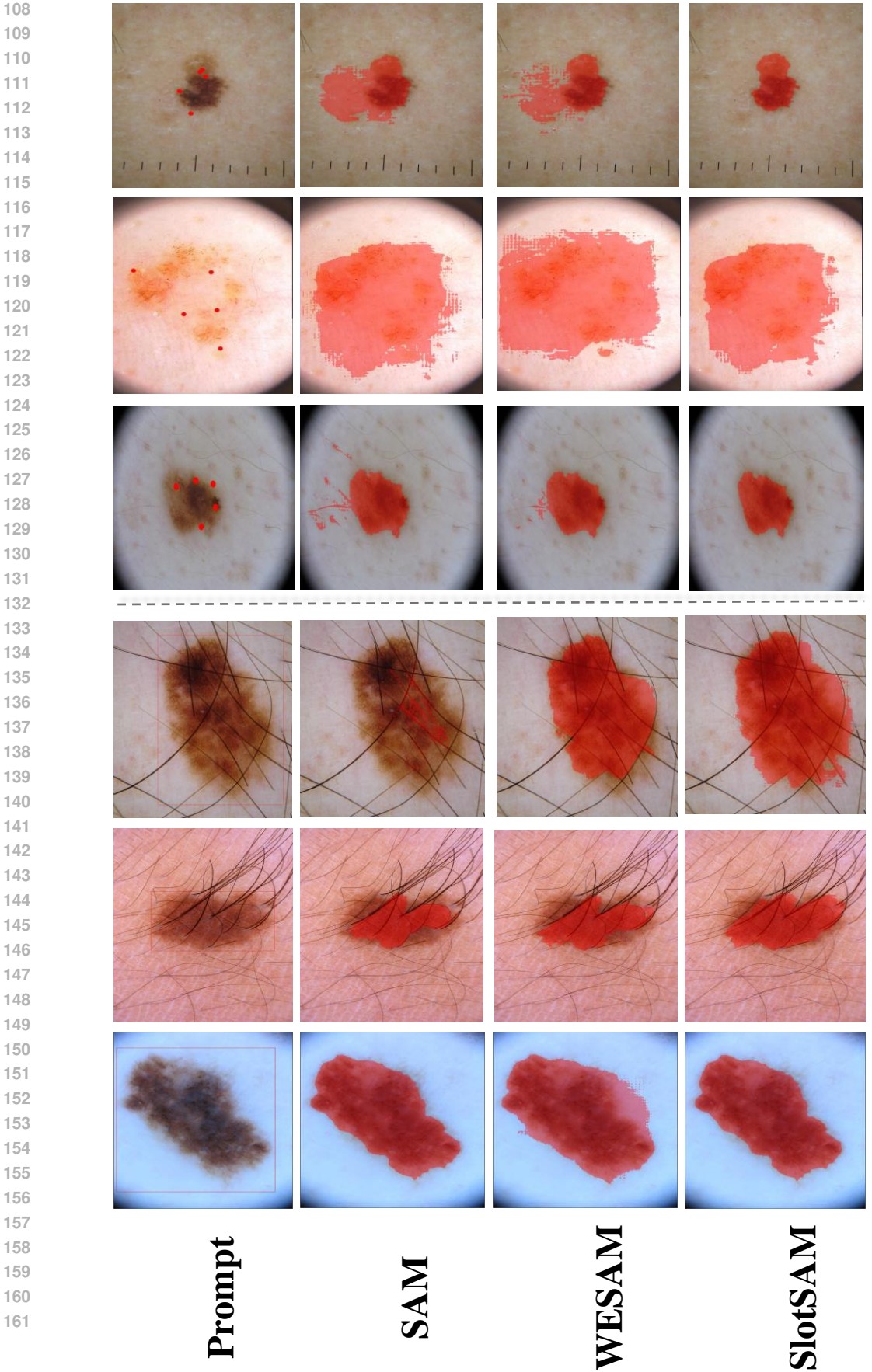

Figure 2: Comparison between SlotSAM and SOTAs of the fineness of the predicted masks on ISIC.

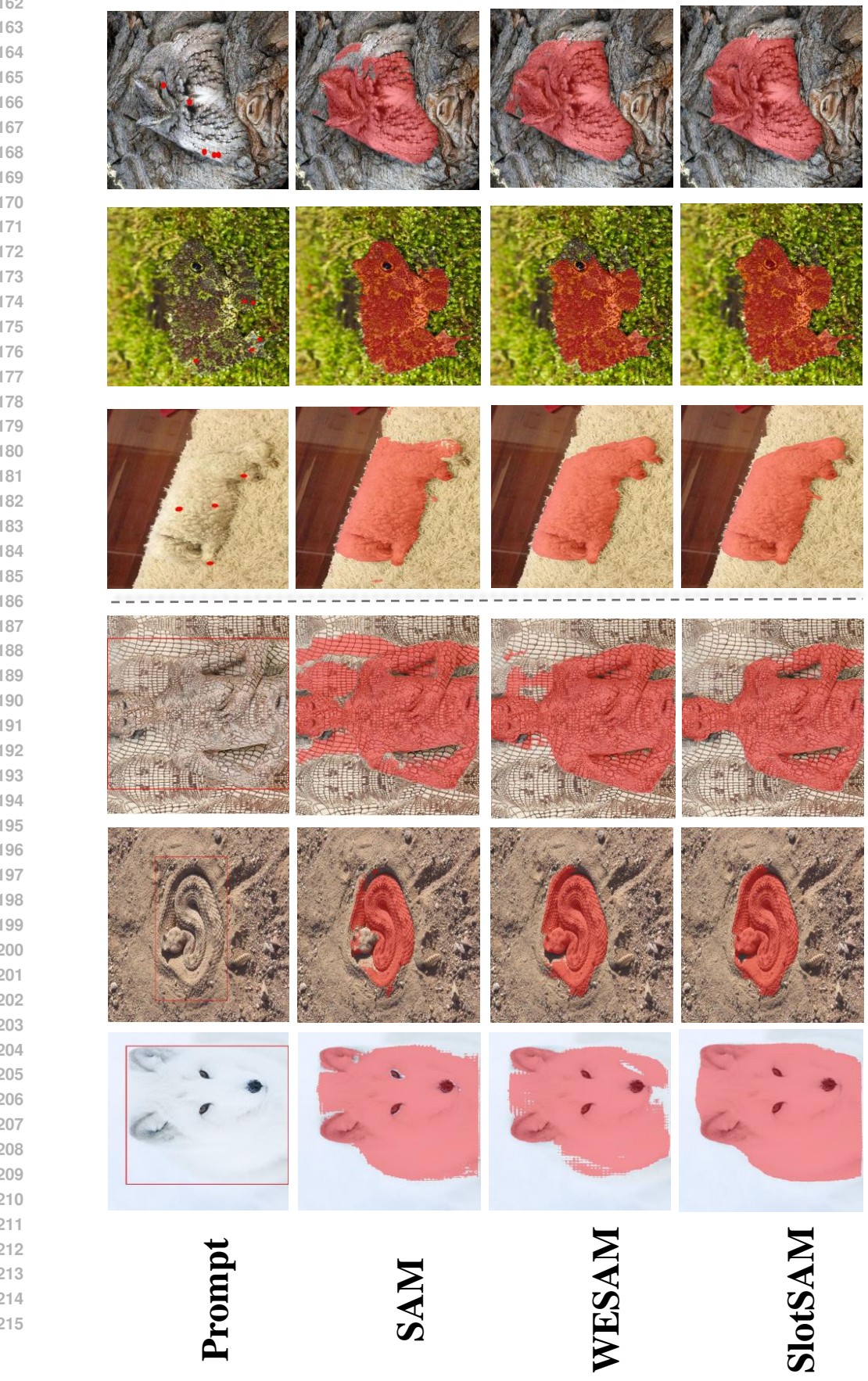

Figure 3: Comparison between SlotSAM and SOTAs of the fineness of the predicted masks on CAMO.

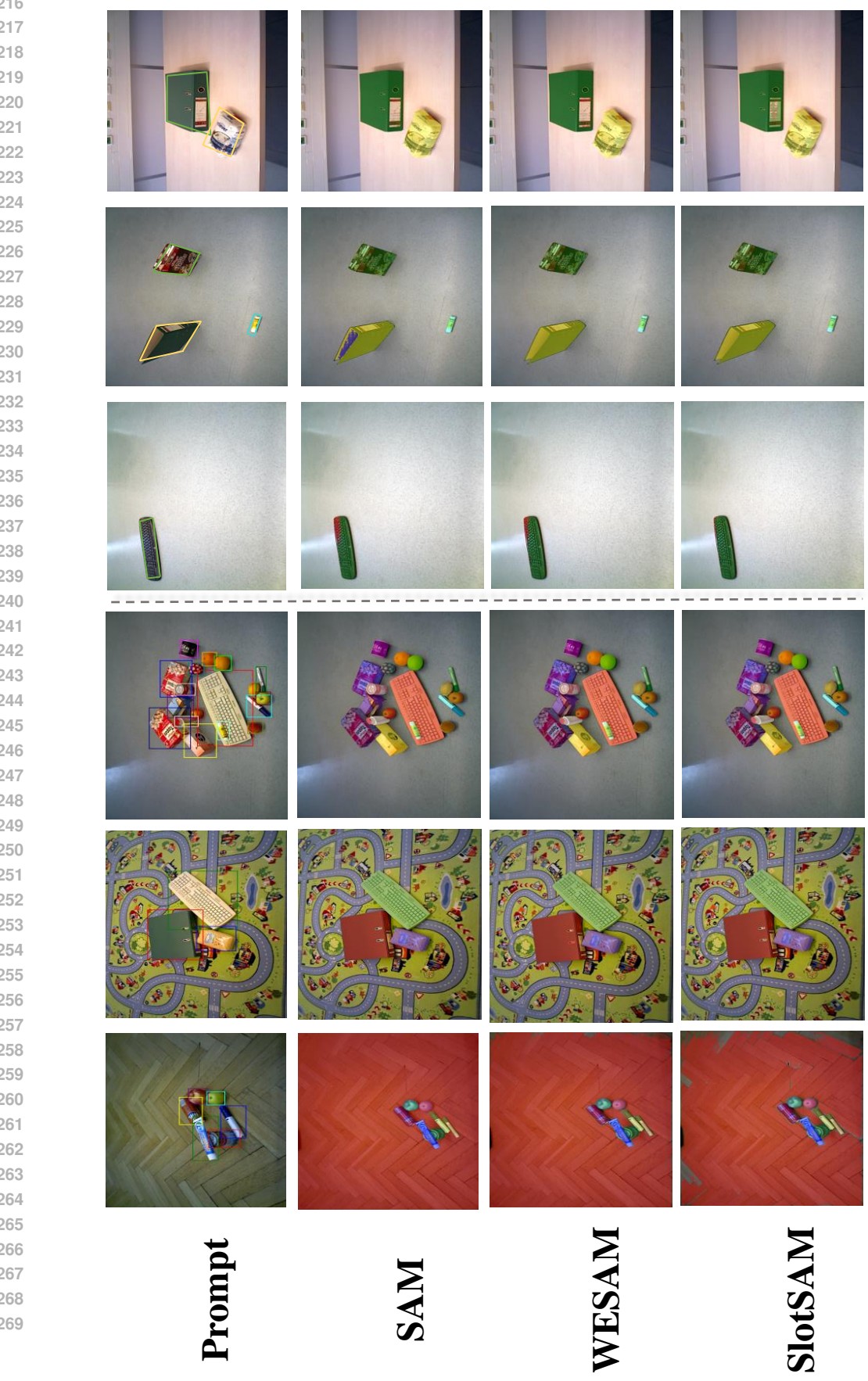

Figure 4: Comparison between SlotSAM and SOTAs of the fineness of the predicted masks on OCID.

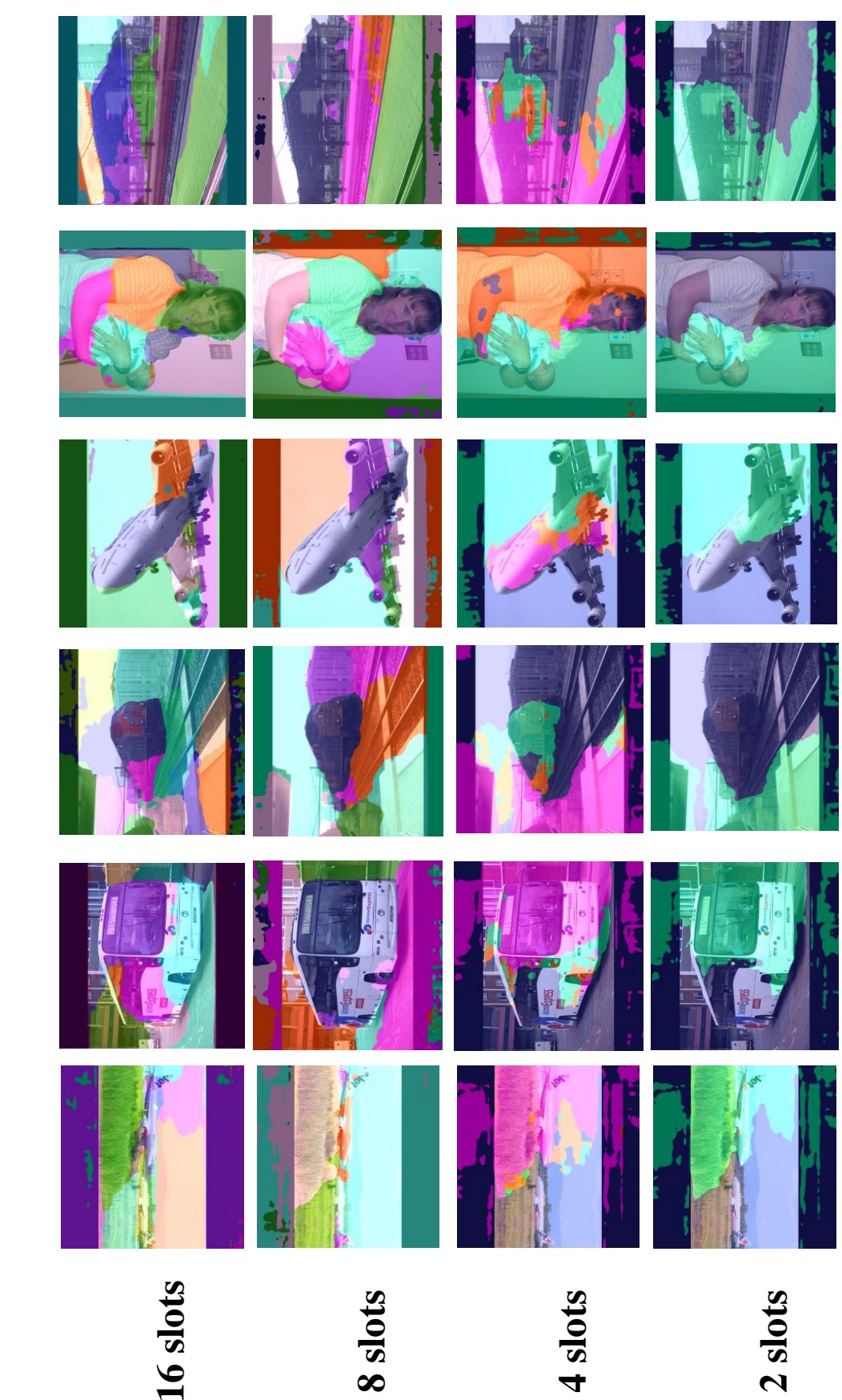

Figure 5: Semantic competition and semantic degradation exists among different numbers of slots on Pascal VOC.

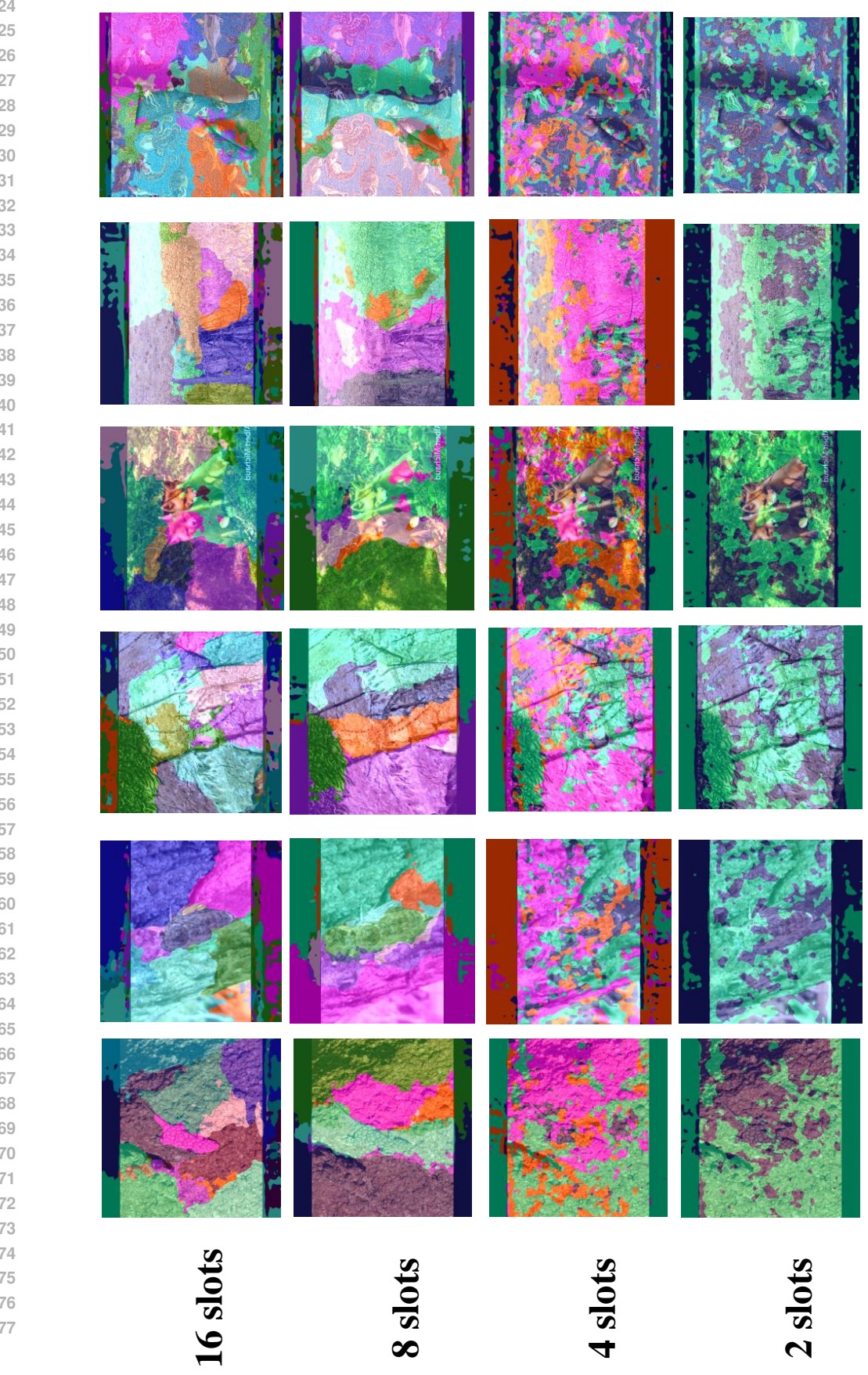

**16 slots**  **8 slots**  **4 slots**  **2 slots**

Figure 6: Semantic competition and semantic degradation exists among different numbers of slots on CAMO.