# OpenReview forum: "SlotSAM: Bootstrap Segmentation Foundation Model under Real-world Shifts via Object-Centric Learning"
_ICLR.cc/2025/Conference — ICLR 2025 Conference Withdrawn Submission_

### Official Review · Reviewer_Vxxm · 2024-10-31

**Soundness:** 3
**Presentation:** 2
**Contribution:** 2
**Rating:** 5
**Confidence:** 3

**Summary:**

The paper introduce SlotSAM, a method that reconstructs features from the encoder in a self-supervised manner to create object-centric representations. These representations are then integrated into the foundation model, bolstering its object-level perceptual capabilities while reducing the impact of distribution shifts. Extensive experiments is conducted to demonstrate the effectiveness of SlotSAM.

**Strengths:**

The idea of obtaining high-quality representations from foundation model and projecting them as object tokens is well-motivated.
The architecture of the proposed SlotSAM seems reasonable. Experiments are also extensive.

**Weaknesses:**

1. The writing needs to be improved. There are many unclarities in this paper. For example,
(1) In Line126-134, many notations are not explained.

(2) In Line 128, what is the difference between strong augmentation and weak augmentation? The role of anchor model, student model and teacher model also need to be explained.

(3) In Line 150, the linear transformation $\mathcal{K}_\beta$ is for slots, but why it is used to process $\mathbf{z}$ in Line 153. In Line 154, why the query is a function of the slots?

(4) In Line 156, what is the architecture of slot-decoder?

(5) In Line 188, how to perform ''nonlinear combination'' ?

You may consider adding a notation table or glossary to address point (1), and perhaps dedicate a subsection or appendix to explain the concepts in points (2)-(5).

2. In Table 1, SlotSAM outperforms fine-grained mask supervised fine-tuning on some datasets but not on others. Are there any insights into this discrepancy? Are the potential factors contributing to this discrepancy, such as dataset characteristics or task-specific challenges?

Though the idea is well-motivated and experiments are extensive, the presentation in Section 2-3 is quite poor.  If the author solves my problem, I will consider raising my score.

**Questions:**

See Weaknesses.

---

### Official Review · Reviewer_Kn8G · 2024-11-03

**Soundness:** 2
**Presentation:** 2
**Contribution:** 3
**Rating:** 5
**Confidence:** 4

**Summary:**

This work identifies and addresses the issue of real-world shift, which includes distribution shift and prompt shift, challenges that hinder the performance of the Segment Anything Model (SAM) and Foundation Models in general. The authors introduce SlotSAM, a method that builds on SAM by incorporating object-centric learning. SlotSAM is aimed at mimicing human-like scene decomposition, enabling the model to distinguish and represent objects more accurately. SlotSAM leverages pre-trained foundation models and employs self-supervised learning to create object tokens, which are then injected into SAM’s existing architecture. The authors demonstrate that SlotSAM achieves improved segmentation performance on benchmark datasets, handling out-of-distribution data and varying prompt types.

**Strengths:**

- The work introduces a novel approach to enhancing segmentation foundation models by applying object-centric learning principles
- The idea is very interesting in mimicking human like behavior of scene decomposition into different object tokens which can be leveraged downstream
- The results seem promising

**Weaknesses:**

- The paper discusses Segmentation Foundation Models, but is only demonstrated with SAM. It is also mentioned that it easy to apply to other models yet this is not elaborated
- **Section 3 Methodology** is poorly written. The method can be better explained in text, as well as having a more representative architecture diagram possibly also illustrating the training regime. It was not an easy read trying to understand what is going on, as SlotSAM introduces a fairly elaborate training procedure
- Most figures (especially Figure 7) are not well presented, or unintuitive. They also lack proper captions that detail what is going on in the figure
- The prompt shift formulation is unclear, and not well motivated

**Questions:**

- Will the code be made public ?
- Please address the weaknesses which are highlighted. Especially the method description and training procedure

---

### Official Review · Reviewer_6XjF · 2024-11-04

**Soundness:** 2
**Presentation:** 3
**Contribution:** 2
**Rating:** 5
**Confidence:** 3

**Summary:**

In this paper, a new objective is defined for reconstructing the slot-attention as high-level features with stronger inductive biases.  Since the encoder of the foundation model effectively extracts high-level semantics for each object within the image, it offers a uniform representation of the high-dimensional nature of the real world without being biased by pixel color reconstruction. After the acquisition of high-quality object-centric representations, they could be seamlessly integrated with existing tokens in most foundation models and can be considered as object tokens. During the forward process, object tokens can leverage the attention mechanism among tokens to access global image context, geometric region, semantic information, and mask regions. The overall idea is interesting. However, there are several points to be addressed.

**Strengths:**

1. By integrating slot-attention with high-level semantic features, the model achieves improved generalization across diverse and complex tasks.
2. Well-written, high-quality figures and good logic.

**Weaknesses:**

1. The model's performance shows significant sensitivity to slot count, especially in complex scenarios. This could limit the model's generalization ability across different tasks, as the optimal configuration needs to be manually adjusted based on the task, resulting in a lack of consistency.
2. Although this paper somehow improves the neural network architectures, according to the experiment results. The reason why this architecture works is not clear. Besides, is this the only way to modify the neural network architecture. For example, what if using other kinds of attention mechanism, etc.  What is the most important reason why this architecture works in fine-tuning? Is it possible to achieve the same improvement with minor architecture change given the SAM is already trained with a lot of data.
3. Why modifying neural architectures can lead to object-centric representations?

    In this paper, it says “The underlying logic of Slot-Attention is to reconstruct features through self-supervision, compressing high-dimensional, semantically rich, and unstructured object features into low-dimensional structured information in a bottleneck-like manner.”

    Why the reconstruction can lead to object-centric representations. Is there any theoretical or experimental evidence for showing this.

    Indeed, information bottleneck method is a way for compressing information. But why the compressed information will be object-centric. Will any method for information compressing can be object-centric. This seems to be a bit too vague.
4. The same applies for the object-centric representation injection part of this paper.

**Questions:**

1. The paper lacks detailed experiments and validation on how effectively object tokens interact with the base model’s tokens.
2. Slot-Attention updates each slot’s representation iteratively through GRU. Why choose GRU, have you considered other updade mechanisms such as MLP? How is the number of GRU iterations determined in multi-object scenarios? Can you conduct an ablation study with different iteration counts?
3. The method applies the slot mechanism after the entire encoder. Would it be feasible to integrate the slot mechanism into intermediate outputs within the encoder?
4. For the trade-off in the number of slots: does a higher slot count reduce training speed, and could increased competition between slots actually lower model performance?

---

### Official Review · Reviewer_j8rk · 2024-11-05

**Soundness:** 1
**Presentation:** 1
**Contribution:** 2
**Rating:** 3
**Confidence:** 4

**Summary:**

The authors examine how current segmentation models like SAM are adapted for new tasks, e.g. with fine-tuning or prompt engineering, yet still struggle with real-world distribution shift and inconsistent prompting strategies during fine-tuning and testing. To address this, they propose to learn object-centric representations from a pretrained encoder (as done in previous work) and then reintroduce them into the segmentation model. The idea is that high-level object-centric features are more robust to distribution shifts and by extension make the resulting method more robust. Tests on datasets spanning natural, medical, and robotic images show the proposed method outperforms existing models, providing more accurate and reliable segmentation with minimal supervision.

**Strengths:**

The topic is relevant for the conference, the proposed method is sensible and interesting, and it appears to improve over the considered baselines.

**Weaknesses:**

Although somewhat incremental, the paper is overall interesting and relevant. Unfortunately:
- The paper is challenging to read in its current form, with the proposed method explained in a way that feels overly convoluted and unclear.
- The context in which this work is situated isn’t clearly established, and is at times misleading.
- The method is presented as broadly applicable to vision foundation models, but the focus is limited to SAM, and the experiments don’t fully match the broader scope suggested in the introduction.

## More detailed comments.

**Motivation and context.**
- In discussing the motivation (lines 72-77), I found the comments on Slot Attention quite misleading. The 3 points the authors mention are in essence the story up to 2022 or so. Then DINOSAUR and other works came about. Very few of these are mentioned in the paper, and those that are, are only mentioned later.
- Similarly, the statement "our objective is to redefine the reconstruction target of Slot-Attention as high-level features with stronger inductive biases" also seems misleading. This has been explored in the previous work I was referring to above (e.g., DINOSAUR). If the proposed method differs, it would help to clarify what’s unique in this approach. Some relevant work appears later, but mentioning it upfront would improve clarity.
- It’s also unclear why SAM was chosen specifically as a "representative foundation model". Could this method apply to other pretrained vision models like DINO, MAE, or JEPA? Has this been investigated or what is the argument for not investigating it? I would expect the authors to resolve this mismatch, e.g. by focusing the entire story on SAM or at least segmentation models, or alternatively including other foundation models.
- The related work, especially on object-centric learning methods from the last couple of years, is quite limited.

**Clarity.**
Generally, the context and motivation could be clearer in terms of language, too. The proposed model is not overly complex, but the explanation feels unnecessarily convoluted. A simpler, more straightforward presentation would help. Here are a few specific examples:
- Some parts of the background/related work section are confusing. For example, “Recent research (Li et al., 2024) has employed Stable-Diffusion to enhance a subset of the SA-1B (Kirillov et al., 2023) dataset, which requires unsustainable consumption of resources” – the purpose of enhancing the dataset isn’t clear. More context here would improve understanding.
- Terms like "prompt shift" might not be clear to readers unfamiliar with VLMs; for instance, “where downstream tasks provide only coarse weak supervision instead of the fine-grained labels available in the source domain” could use more background.
- "it offers a uniform representation of the high-dimensional nature of the real world without being biased by pixel color reconstruction": here “uniform representation” doesn’t clearly convey the intended meaning.
- "we innovatively design the object-centric representation stored in the slots to be the Object Token": Terms like "object token" should also be introduced in a more intuitive way for readers unfamiliar with this terminology.
- "object-centric learning reduces reliance on domain-specific variables and enables more robust handling of out-of-distribution data": this is a pretty strong claim which may require e.g. some more explanation and/or references. Ideally, one would run an ablation study with alternative representation mechanisms that do not include slot-based approaches.
- Some sentences could be rephrased for clarity, like “To avoid disrupting the optimization preference established by the decoder for existing tokens.”
- For readers not familiar with SAM, it may be unclear what "prompts" are in this context. In addition, concepts like point prompts should be introduced earlier and with more context.

**Experiments.**
- The results in Table 2 could be made clearer. For instance, the $\Delta$ row reflects improvement over SAM, but in some cases, improvements over WESAM are minimal. Including deltas for all methods or a bar plot would make the comparisons easier to interpret.
- Some discussion on SAM’s different prompt types and their expected impact on performance would be helpful.
- It would also help to explain degrees of weak supervision in more detail, along with any variations in supervision “strength” and how that might influence results.
- As mentioned earlier, an ablation replacing slot attention with other (simpler) approaches could be interesting. On the other hand, some of the current ablation studies focus on aspects that can almost be considered implementation details and do not necessarily provide interesting insights.

**Minor.**
There are a few typos and sentences that need adjustment. Just a few examples:
  - For instance, in the first paragraph of the introduction: “Despite SAM’s claims robust zero-shot segmentation capabilities” could be “Despite claims of SAM’s robust zero-shot segmentation capabilities” or something to that effect.
  - Shortly after, “Enhancing SAM’s generalization and robustness on new data is a key focus”: a focus of what, or by whom?
  - Line 151 includes a repetition: “Slot-Attention is trained as $\mathrm{update}(\mathbf{A},\mathbf{v}) = \mathbf{A}^T \mathbf{v}$, where $\mathrm{update}(\mathbf{A},\mathbf{v}) = \mathbf{A}^T \mathbf{v}$”


## Conclusion

The idea is interesting and worth exploring further, but several major issues prevent me from recommending acceptance at this stage: the overall clarity of the exposition, particularly regarding the proposed method; the accurate positioning of this work within the existing literature; and, while the experiments are detailed, they don’t fully align with the broader scope the paper initially suggests.

**Questions:**

See weaknesses.

---

### Note · Authors · 2024-11-20

**Comment:**

We would like to express our sincere gratitude to the Reviewers and Chairs for their invaluable feedback and insights. After careful consideration, we decide to withdraw our submission to further develop and refine our paper based on these constructive comments. Thank you once again for your time and thoughtful evaluation.

**Withdrawal Confirmation:**

I have read and agree with the venue's withdrawal policy on behalf of myself and my co-authors.